# Voxel Transformer with Density-Aware Deformable Attention for 3D Object Detection

**DOI:** 10.3390/s23167217

**Published:** 2023-08-17

**Authors:** Taeho Kim, Joohee Kim

**Affiliations:** Department of Electrical and Computer Engineering, Illinois Institute of Technology, Chicago, IL 60616, USA; tkim46@hawk.iit.edu

**Keywords:** 3D object detection, transformer, deformable attention, density, deep learning

## Abstract

The Voxel Transformer (VoTr) is a prominent model in the field of 3D object detection, employing a transformer-based architecture to comprehend long-range voxel relationships through self-attention. However, despite its expanded receptive field, VoTr’s flexibility is constrained by its predefined receptive field. In this paper, we present a Voxel Transformer with Density-Aware Deformable Attention (VoTr-DADA), a novel approach to 3D object detection. VoTr-DADA leverages density-guided deformable attention for a more adaptable receptive field. It efficiently identifies key areas in the input using density features, combining the strengths of both VoTr and Deformable Attention. We introduce the Density-Aware Deformable Attention (DADA) module, which is specifically designed to focus on these crucial areas while adaptively extracting more informative features. Experimental results on the KITTI dataset and the Waymo Open dataset show that our proposed method outperforms the baseline VoTr model in 3D object detection while maintaining a fast inference speed.

## 1. Introduction

Three-dimensional object detection from point clouds, which is crucial for autonomous driving, has seen significant growth. Compared to CNNs, transformer-based models are superior in handling long-range dependencies due to their expansive receptive fields. However, the transformer models require high computational costs and slower convergence. Additionally, the effectiveness of hand-crafted attention patterns can be compromised, as they may miss relevant keys/values. In this paper, we introduces the Voxel Transformer with Density-Aware Deformable Attention (VoTr-DADA). This innovative framework enhances detection performance by effectively addressing these issues.

Numerous studies [1,2,3,4,5,6,7,8,9,10,11,12] have developed CNN-based models for this task. Recent works [6,7,10] have utilized 3D sparse CNN for efficient feature extraction. Despite the success of CNN-based models in various computer vision tasks, their direct application to 3D object detection is challenging. This is primarily due to the sparsity and irregular distribution of 3D point clouds in continuous space, which complicates the efficient use of CNN layers that are designed for regular data structures. CNN-based methods require the stacking of multiple convolutional layers to capture broader contextual information, due to their fixed and limited receptive field, which significantly escalates computational costs.

Transformer-based models, initially designed for neural language processing (NLP) tasks, have recently become popular in object detection [13,14,15,16,17,18,19,20,21]. Vision Transformer (ViT) [18], the first transformer model applied to vision tasks, has significantly influenced the field of computer vision with its attention-centric approach. This transformer utilizes a self-attention mechanism that is invariant to the input element permutation and cardinality. This distinctive feature makes the transformer especially effective for processing point cloud data, as it can adeptly capture long-range dependencies and contextual relationships among points. This transformer has demonstrated superior performance over traditional CNN models in tasks such as image classification, detection, and segmentation. DETR [13] pioneered the use of transformers for 2D object detection through direct set prediction, and 3DETR [21] extended this approach to 3D object detection, directly attending to points in 3D space.

A Voxel Transformer (VoTr) [15], a transformer-based 3D object detection model, effectively captures long-range voxel relationships through self-attention without significantly increasing computational demands. Despite its strengths, VoTr struggles with detecting distant objects with fewer points, mainly due to its use of inflexible pre-defined receptive fields. Inspired by deformable convolution [20] and deformable attention [22], a Deformable Attention Transformer (DAT) [19] extends existing attention modules by integrating deformable features. It utilizes a lightweight CNN offset network to shift keys and values to important regions, reducing computational overhead. However, implementing deformable attention in 3D object detection is challenging due to the high computational demand of using a 3D CNN for offset learning.

Recently, many studies [23,24,25,26] have explored various approaches to leverage density features for 3D object detection. Density-Aware PointRCNN [23] and Density Awareness and Neighborhood Attention [24] have shown how adjusting voxel size based on point density or using density-aware features can enhance object detection in LiDAR-based systems. Similarly, DA-3DSSD [25] proposes a density-aware single-stage 3D object detector that dynamically modifies its receptive fields to capture the features of objects with varying densities efficiently. In P-DAV [26], point density is exploited to adjust voxel size while preserving the geometric structure of the point cloud. In this study, we leverage these density-focused insights to identify significant voxels for deformable attention, thereby improving the performance of our 3D object detection model.

In this paper, we propose a 3D object detection backbone called Voxel Transformer with density-aware deformable attention (VoTr-DADA), which improves VoTr [15] by introducing a simple voxel density-based deformable attention mechanism. The proposed density-aware deformable attention (DADA) module is designed to compute deformable attention efficiently so that it can focus on more informative regions and capture long-range relationships among non-empty voxels. Specifically, after selecting the attending voxels for a querying voxel using local and dilated attentions, as in VoTr, the proposed DADA module shifts the attending voxels to the locations with higher voxel densities. Differently from existing methods [19,20,22], which use a lightweight CNN to learn the offsets for deformable attention, the proposed method exploits the voxel density features of point clouds to enlarge the attention range in a simple and more flexible way. Experimental results on KITTI and Waymo Open datasets show that the proposed method outperforms the vanilla VoTr in 3D object detection while maintaining a comparable computational cost and faster processing speed. The main contributions of the proposed method are as follows:We propose a 3D object detection backbone architecture, VoTr-DADA, which improves VoTr [15] for 3D object detection. The proposed density-aware deformable attention module exploits voxel densities to direct the attending voxels to more informative regions;We conducted experiments on the KITTI and Waymo Open datasets and show that the proposed VoTr-DADA outperforms the vanilla VoTr with lower computational costs and comparable performance to other transformer-based 3D object detectors.

The rest of the paper is organized as follows. In Section 2, we discuss related work. Section 3 explains the overall 3D object detection architecture and our proposed methods in detail. We present the ablation study along with corresponding experimental results in Section 4. Lastly, we conclude our work in Section 5.

## 2. Related Work

### 2.1. 3D Object Detection

Generally, 3D object detection methods are divided into point-based [2,3,4,11] and grid-based [5,7,16,19] approaches, depending on their data representation. Point-based methods operate directly on raw point clouds to generate 3D boxes. F-PointNet [2] uses frustums of points and applies PointNet [11] for proposal generation, and PointRCNN [3] employs a two-stage pipeline for 3D object prediction. On the other hand, voxel-based methods convert point clouds into regular voxel grids before applying convolutional networks for 3D proposal generation. VoxelNet [5] uses a 3D CNN to extract voxel features from a dense grid, while PV-RCNN [4] introduces a point-voxel-based set abstraction layer using keypoints, thus combining the benefits of both point- and voxel-based approaches. For efficient processing, 3D sparse CNNs [19] are used to avoid the high computational overheads of traditional 3D CNNs. Additionally, to reduce processing costs, LiDAR point clouds are often projected onto a Bird’s Eye View (BEV) map. However, this projection process can result in the loss of crucial 3D information. To address this, several techniques [7,16] compute voxel features using sparse convolutions, project them onto a BEV map, and then predict 3D bounding boxes within this space.

The transformer [27] has made a significant impact on 3D object detection, but applying attention mechanisms efficiently to sparse point clouds remains a challenge. SST [17] addresses this by batching and padding regions with similar token numbers for parallel computation, while VoxSet [14] uses trainable latent codes to connect voxels within each window. Voxel Transformer (VoTr) [15] proposes local and dilated attention mechanisms to compute self-attention on sparse voxels. However, despite the expanded receptive field of VoTr, its flexibility is limited due to the constraints ofn its predefined receptive field. In this paper, we extend the work of VoTr by introducing a voxel density-based deformable attention mechanism to enlarge the attention range in a simple and flexible way.

### 2.2. Deformable Attention

Deformable convolution, a technique adapting to spatial data variations, has been extensively explored [19,20,22] to create flexible receptive fields. Deformable Convolution Networks (DCNs) [20] employ dynamic kernels to capture detailed object features efficiently. This approach has been recently extended to Vision Transformers [19,22]. Deformable DETR [22] combines the benefits of the sparse spatial sampling of deformable convolution with the relational modeling capability of Transformers. The Deformable Attention Transformer (DAT) [19] uses a unique deformable self-attention module that dynamically adjusts key-value pairs based on data, outperforming conventional vision transformer models in image classification and prediction tasks. However, implementing deformable attention in 3D object detection is challenging due to the high computational demand of using a 3D CNN for offset features. To address this, we propose a voxel-based 3D Transformer backbone with deformable attention that efficiently broadens receptive fields without a substantial increase in computational complexity.

### 2.3. Density in 3D Object Detection

Recent studies [23,24,25,26] have highlighted the significance of incorporating density features in 3D object detection tasks. For example, density-aware PointRCNN [23] introduces density-adaptive voxelization (DAV), which adjusts the voxel size based on point density, thereby effectively capturing object features in regions with varying densities. Similarly, Ref. [24] proposes a module called density-aware neighborhood attention (DA-NAM) which utilizes both local and global density-aware features to efficiently distinguish objects from the background in LiDAR-based 3D object detection. Furthermore, DA-3DSSD [25] introduces a density-aware single-stage detector that uses a density-aware module to dynamically adjust the detector’s receptive field. This adjustment allows the detector to capture features of objects with different densities more effectively. P-DAV [26] employs point density-aware voxels to enhance LiDAR-based 3D object detection performance. It dynamically adjusts the voxel size based on point density, resulting in better preservation of the geometric structure of the point cloud. Our proposed method incorporates density features to identify the significant voxels for deformable attention. This enables us to select deformed attending voxels for multi-head self-attention and enhances the overall effectiveness of our 3D object detection model.

## 3. Proposed Method

In this section, we present VoTr-DADA, a voxel-based transformer backbone specifically designed for 3D object detection from point clouds. VoTr-DADA improves upon the Voxel Transformer (VoTr) [15] by integrating a Density-Aware Deformable Attention (DADA) module. The following sections provide a detailed description of our proposed method.

### 3.1. Preliminaries

Voxel Transformer (VoTr) [15] is a voxel-based transformer backbone for 3D object detection from point clouds. This model utilizes sparse and submanifold voxel modules to efficiently capture long-range context information from sparse point clouds using attention mechanisms. The sparse voxel module can extract features from the empty locations and, therefore, expand the non-empty space. On the other hand, the submanifold voxel module is strictly applied to the non-empty voxels to maintain the original 3D structure of the input point cloud. VoTr proposes two attention mechanisms, called local attention and dilated attention, to determine the attending voxels for a querying voxel in multi-head attention. Specifically, local attention focuses on the neighboring region around the querying voxel to preserve local details, while dilated attention is designed to capture broader context information.

### 3.2. Overall Architecture

Figure 1 shows the architecture of the proposed VoTr-DADA backbone for 3D object detection. The proposed VoTr-DADA enhances the original VoTr by introducing the density-aware deformable attention (DADA) module, which enables it to compute self-attention more efficiently without increasing the computational complexity significantly. The proposed VoTr-DADA 3D backbone consists of three building blocks: one VoTr block and two VoTr-DADA blocks. Specifically, the VoTr block is composed of one sparse voxel module and two submanifold voxel modules, as in the vanilla VoTr. In the VoTr-DADA block, we replace the submanifold voxel module with our DADA module. Therefore, a VoTr-DADA block consists of one sparse voxel module and two DADA modules. With its flexible receptive fields, VoTr-DADA greatly enhances global context modeling, surpassing the performance of the original VoTr architecture. The 3D backbone architecture for our proposed model consists of one VoTr block followed by two VoTr-DADA blocks, which was determined based on extensive experiments and ablation studies.

The input point cloud is first partitioned into sparse voxel grids, and the VoTr-DADA is then applied to extract voxel features. In the first VoTr block, a sparse voxel module and two submanifold voxel modules are used, as in the vanilla VoTr, to capture the contextual information in different ranges based on multi-head self-attention. Following the initial VoTr block, we apply our proposed VoTr-DADA blocks. These blocks employ Density-Aware Deformable Attention (DADA) modules to deform attending voxels from the previous block. The voxel features extracted by the VoTr-DADA 3D backbone are then projected onto a bird’s eye view (BEV) feature map for generating 3D proposals, followed by region-of-interest (ROI) refinement.

### 3.3. Density-Aware Deformable Attention Module

As illustrated in Figure 2, the DADA module utilizes density features to compute data-dependent sparse attention. Instead of relying on the convolutional networks to learn offsets, it identifies important regions based on voxel density. This approach reduces computational complexity while maintaining performance. The DADA module uses a voxel density function to locate the voxel with the highest density near the attending voxels.

We divide the 3D space of the point cloud into equally spaced voxels. The dimensions of the 3D space (X,Y,Z) are denoted as *H*, *W*, and *D*, respectively. Each voxel has a size of $V_{H}$, $V_{W}$, and $V_{D}$ in the corresponding dimensions. We use a dense voxel grid to rasterize the whole 3D space and only maintain the non-empty voxels in an array V and corresponding feature array F. Following the VoTr [15], we determine the attention range $\Omega (v_{i})\subseteq V$ and attending voxels $v_{j}\in \Omega \left(v_{i}\right)$ for a given querying voxel $v_{i}$ using local and dilated attentions. To efficiently find the highest-density voxel near the attending voxels, we define a search range $\Delta (v_{j})$ using the coordinates and the size of the attending voxels. The search range $\Delta (v_{j})$ can be represented as follows:(1)$$\Delta \left(v_{j}\right)=\left\{\left(x,y,z\right)\in R^{3}|\left|x-x_{j}\right|<r\cdot V_{H},\left|y-y_{j}\right|<r\cdot V_{W},\left|z-z_{j}\right|<r\cdot V_{D}\right\},$$
where $\left(x_{j},y_{j},z_{j}\right)\in R^{3}$ denotes the coordinates of the attending voxels, and *r* is the search range parameter. Based on our ablation studies, we set *r* to 4 for the best performance.

To find the highest-density voxel $v_{den}$, we propose a density function H and operate it on the search range $\Delta (v_{j})$ for each attending voxel. The density function H contains two search levels. At each level, we utilize an octree space subdivision method [28] to divide the space into eight equal-sized volumes. The size of a volume at each search level can be expressed as
(2)$$V_{H}^{l}=\frac{r\cdot V_{H}}{2^{l}},\;V_{W}^{l}=\frac{r\cdot V_{W}}{2^{l}},\;V_{D}^{l}=\frac{r\cdot V_{D}}{2^{l}},$$
where *l* = 1, 2 is the search level, and $V_{H}^{l}$, $V_{W}^{l}$, and $V_{D}^{l}$ are the height, width, and depth of the volume at level *l*, respectively. For each attending voxel, the voxel density function initializes a 3D volume that matches the size of the entire search range. At level 1, the 3D search range is divided into eight equal-sized volumes, and each has a size of $V_{H}^{1}$ × $V_{W}^{1}$ × $V_{D}^{1}$. Among these subdivided volumes, we search for the highest-density volume depending on the number of points. Then, this highest-density volume proceeds to the next level. We repeat this processing until we find the highest-density voxel $v_{den}$. The proposed voxel density function H can be summarized as follows:(3)$$H\left(\Delta \left(v_{j}\right)\right)=\underset{\begin{array}{c} vol_{n}^{l}\in \Delta \left(v_{j}\right)\;n=1,\ldots ,8 \end{array}}{arg max}\frac{p_{n}^{l}}{V_{H}^{l}\cdot V_{W}^{l}\cdot V_{D}^{l}},\;\mathrm{for}\;l=1,2,$$
where $vol_{n}^{l}$ represents the *n*-th volume at level *l*, and $p_{n}^{l}$ denotes the number of points in $vol_{n}^{l}$. Then, we compare the densities of the highest-density voxel $v_{den}$ and the attending voxel $v_{j}$. The higher-density voxel is used to form the deformed attending voxels $v_{df}$, from which we extract the corresponding deformed keys and values.

Let $f_{i},f_{df}\in F$ be the features of the querying voxel and deformed attending voxels, respectively. We compute the query embedding $Q_{i}$, deformed key embedding $K_{df}$, and deformed value embedding $V_{df}$ as
(4)$$Q_{i}=f_{i}W_{q},\;K_{df}=f_{df}W_{k}+E_{pos},\;V_{df}=f_{df}W_{v}+E_{pos},$$
where $W_{q},W_{k},$ and $W_{v}$ are the linear projection of query, key, and value, respectively, and the deformed positional encoding $E_{pos}$ can be calculated by
(5)$$E_{pos}=\left(C_{i}-C_{df}\right)W_{pos},$$
where $C_{i}$ and $C_{df}$ are the center of the querying voxel and the deformed attending voxels, and $W_{pos}$ is the linear projection of relative position. Thus, self-attention on deformed attending voxels can be formulated as
(6)$$f_{df}^{attention}=\sum_{v_{df}\in \Delta \left(v_{j}\right)}\sigma \left(\frac{Q_{i}K_{df}}{\sqrt{d}}\right)\cdot V_{df},$$
where σ(·) is the softmax normalization function and *d* is the embedding dimension.

## 4. Experimental Results

In this section, we demonstrate that VoTr-DADA achieves comparable performance to recent state-of-the-art 3D detection models, including the vanilla VoTr, while maintaining lower computational complexity. We provide implementation details of the proposed architecture and present ablation results to validate the effectiveness of our model. Our evaluation is conducted on the KITTI and the Waymo Open datasets.

### 4.1. Dataset and Evaluation Metric

**Dataset.** We evaluated VoTr-DADA on the Waymo Open dataset [29] and the KITTI dataset [30]. The KITTI dataset includes 7481 training images/point clouds and 7518 test images/point clouds, covering three categories: car, pedestrian, and cyclist. The point cloud data are reserved within the range of (0 m, 70.4 m), (−40.0 m, 40.0 m), and (−3.0 m, 1.0 m) on the x, y, and *z* axes, respectively. The training samples are divided into 3712 train and 3769 validation splits. The Waymo Open Dataset (WOD) contains 1150 sequences in total (more than 200 K frames): 798 for training, 202 for validation, and 150 for testing. Point cloud sequences cover a perception range of 150 m × 150 m.

**Metric.** The official evaluation metric on the KITTI dataset is 3D mean average precision (mAP), with a rotated IoU threshold 0.7 for cars. On the *test* set, mAP was calculated with 40 recall positions by the official server. The results on the *val* set were calculated with 11 recall positions for a fair comparison with other approaches. Performance was measured based on different difficulty levels (easy, moderate, difficult). The results are reported herein based on object size, occlusion condition, and truncation level. We followed the official metrics on Waymo to calculate the standard 3D mean average precision (mAP) metric and heading-weighted 3D mAP (mAPH) of all methods. Both metrics were based on an IoU threshold of 0.7 for vehicles and 0.5 for other categories. Performance was measured based on distance ranges (0–30 m, 30–50 m, over 50 m) and two levels of difficulty: LEVEL 1 (boxes with more than five LiDAR points) and LEVEL 2 (boxes with at least one LiDAR point).

### 4.2. Experimental Setup

**Model.** VoTr-DADA (SSD) is an extension of VoTr-SSD, which is based on the single-stage model SECOND [7]. Similarly, VoTr-DADA (TSD) builds upon VoTr-TSD, which is grounded in the two-stage framework PV-RCNN [4]. In both cases, we replaced the original backbone with our proposed VoTr-DADA. For VoTr-DADA (SSD), we maintained the anchor-based assignment, while, for VoTr-DADA (TSD), we used key points to extract voxel features from the voxel transformer for second-stage RoI refinement.

**Implementation Details.** During training, each querying voxel had a total of 48 attending voxels, and multi-head attention was performed with 4 heads. The initial voxel size was set to (0.05 m, 0.05 m, 0.1 m). After downsampling the voxels with a stride of two through the sparse voxel module, the voxel size became eight times larger than the initial voxel size. Therefore, the first DADA module operated on the voxels that had been downsampled twice, resulting in a voxel size of (0.2 m, 0.2 m, 0.4 m). In dense areas, the number of points ranged from 30 to 50, while, in sparse areas, it was typically less than 5 and could be as low as 1. To ensure a uniform distribution of data, we set the maximum number of points to 10 for voxel size (0.2 m, 0.2 m, 0.4 m) and 80 for voxel size (0.4 m, 0.4 m, 0.8 m), with a normalized density value of 1. Points exceeding the maximum number were not included in the calculation of voxel density, ensuring manageable density levels even in highly concentrated point areas. VoTr-DADA was trained using the ADAM optimizer within the entire framework. On the KITTI dataset, VoTr-DADA (SSD) and VoTr-DADA (TSD) were trained with batch sizes of 32 and 16, respectively. The learning rate was set to 0.01, and the cosine annealing strategy was adopted for the learning rate decay. Training was conducted for 80 epochs using 2 NVIDIA Titan RTX GPUs. On the Waymo Open dataset, 20% of the frames were uniformly sampled for training, and the full validation set was used for evaluation. VoTr-DADA (SSD) and VoTr-DADA (TSD) were trained with a batch size of 16 and a learning rate of 0.003. The training was performed for 60 epochs for VoTr-DADA (SSD) and 80 epochs for VoTr-DADA (TSD) using 2 NVIDIA Titan RTX GPUs. Data augmentation and other configurations were kept consistent with the corresponding baselines.

### 4.3. Experimental Results for 3D Object Detection

**Performance on the KITTI dataset.** As is shown in Table 1, VoTr-DADA (SSD) and VoTr-DADA (TSD) achieved 79.31% mAP and 84.63% mAP on the moderate car class on the KITTI *val* split, respectively. For the hard car class, VoTr-DADA (TSD) achieved 78.90% mAP, outperforming most of the previous methods. This suggests that the long-range voxel relationships captured by VoTr-DADA are crucial for detecting 3D objects with few points. As the Table 2 shows, compared to VoTr-TSD, VoTr-DADA (TSD) improved 0.27% mAP and 0.19% mAP on the moderate and hard car classes on the KITTI *test* set, respectively. The performance of VoTr-DADA at the ’Easy’ level was found to be slightly lower or comparable to other models, while it generally outperformed them at the ’Moderate’ and ’Hard’ levels. This suggests that VoTr-DADA performs exceptionally well in sparse areas. The relatively lower performance at the ’Easy’ level can likely be attributed to information loss that occurs through downsampling in denser areas.

**Performance on the Waymo Open dataset.** As is shown in Table 3, simply switching from the VoTr backbone to VoTr-DADA resulted in a 1.03% and 0.74% mAP improvement for VoTr-SSD and VoTr-TSD, respectively, in the LEVEL 1 category. In the range of 30–50 m and 50 m–Inf, VoTr-DADA (SSD) achieved 1.06% and 1.13% improvements, and VoTr-DADA (TSD) achieved 0.47% and 0.52% improvements on LEVEL 1. These significant performance gains in the distant area demonstrate the capability of VoTr-DADA to capture broader contextual information in multi-head self-attention.

### 4.4. Ablation Study

We conducted ablation studies to verify the effectiveness and characteristics of our processing pipeline. In this section, all the involved model variants were built upon the VoTr-SSD baseline and evaluated on the KITTI dataset using the evaluation metric of average precision, calculated with 40 recall positions.

**Comparison of the inference speed.** Table 4 demonstrates that VoTr-DADA maintained computational efficiency, with a processing speed of 13.32 Hz for the single-stage detector. The substitution of VoTr with VoTr-DADA only added a small latency of around 8 ms per frame. However, VoTr-DADA achieved faster processing speed compared to VoTr with convolution-based deformable attention.

**Comparison of model parameters.** Table 5 shows that replacing the 3D convolution-based deformable attention with density-aware deformable attention resulted in a reduction of 0.3 M model parameters. This reduction is primarily due to the fact that the modules in VoTr-DADA do not increase the parameter count, whereas VoTr with 3D convolution-based offset networks typically has a larger number of parameters.

**Effect of the number of attending voxels.** In Table 6, it can be observed that increasing the number of attending voxels from 32 to 48 resulted in a performance boost of 1.35%. This indicates that involving more attending voxels in multi-head attention allows a voxel to obtain richer context information, leading to improved performance.

**Effects of the different VoTr-DADA block structures.** In Table 7, the performance comparison of different VoTr-DADA block structures is presented. Replacing only the first block with the corresponding DADA block resulted in an $AP_{3D}$ of 77.53%. Subsequently, replacing the first and second blocks improved the performance to 78.88%. The highest performance among the different configurations was achieved by substituting the second and third blocks, resulting in an $AP_{3D}$ of 79.54%. Finally, when all the blocks were replaced, the $AP_{3D}$ obtained was 79.36%, showing a slight decrease compared to replacing the second and third blocks. We suppose that this results from the smaller voxel size in the first DADA block, which may be less effective for extracting global contextual information.

**Effects of the highest-density voxel search range.** Table 8 shows the performance comparison of the search range. With a search range parameter *r* of 2, which involves a single division of the search range, we achieved an $AP_{3D}$ accuracy of 78.87%. When *r* was increased to 4, necessitating 2 divisions of the search range, the performance improved to 79.54%. However, with *r* set to 8, which required 3 divisions of the search range, the $AP_{3D}$ performance declined to 77.61%, compared to the results with *r* set to 2 and 4. We suppose that this decrease is due to the overlap of the deformed attending voxels originating from different attending voxels in larger search ranges, which may hinder the effective extraction of local contextual information.

**Qualitative result.** Figure 3 shows the visualization of detection results using the VoTr-DADA on the KITTI dataset.

## 5. Conclusions

This paper proposes a voxel-based transformer backbone called VoTr-DADA for 3D object detection from point clouds. The proposed method improves the vanilla VoTr by introducing a density-aware deformable attention mechanism that enhances self-attention computation for point clouds without increasing computational complexity. By substituting the submanifold voxel module with the DADA module, VoTr-DADA effectively enhances global context modeling, surpassing the VoTr architecture. The DADA module, a key component of VoTr-DADA, leverages density-based deformable attention for efficient, data-dependent sparse attention computation. This module identifies crucial regions based on voxel density, reducing computational complexity while preserving performance.

Experimental results on the KITTI and Waymo Open datasets have validated the performance of VoTr-DADA. The transition to VoTr-DADA led to significant improvements in the LEVEL 1 mAP for both VoTr-SSD and VoTr-TSD, particularly in distant areas. This improvement demonstrates the efficacy of VoTr-DADA in capturing large-scale contextual information for 3D object detection. Furthermore, ablation studies on the model parameters and inference speed show the effectiveness of VoTr-DADA. While VoTr-DADA operates at a fast inference speed, it does experience some latency. To achieve real-time performance, our future work will focus on exploring more efficient transformer-based architectures for 3D detection.

## Figures and Tables

**Figure 1 sensors-23-07217-f001:**
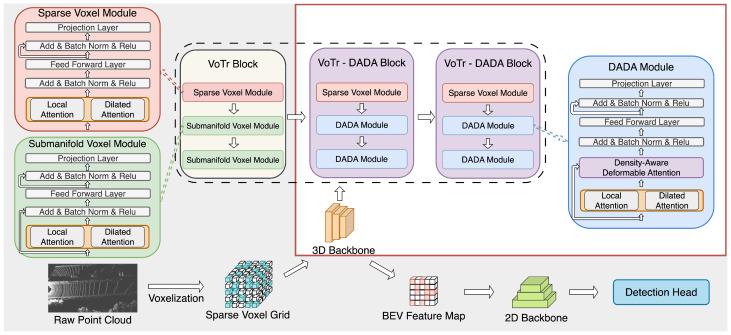
The proposed Voxel Transformer with Density-Aware Deformable Attention (VoTr-DADA) has the same architecture as the voxel transformer (inside the pink-shaded area). The difference is the 3D backbone portion (shown inside the red box). In the DADA module, data-dependent sparse attention is computed using a voxel density-based deformable attention mechanism.

**Figure 2 sensors-23-07217-f002:**
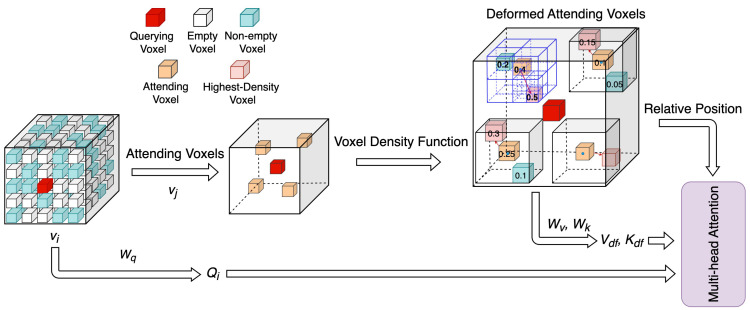
Illustration of the density-aware deformable attention module. The DADA module compares the density of non-empty voxels (cyan) to find the highest-density voxel (pink) for each attending voxel (orange) given a querying voxel (red). The voxel density function efficiently finds the highest-density voxel using octree subdivision (highlighted in blue in the deformed attending voxels). After comparing the density of the highest-density voxel (pink) and the attending voxel (orange), the deformed attending voxel is obtained. The deformed keys and values are then projected from the deformed attending voxels. Additionally, the relative position is computed based on these deformed attending voxels.

**Figure 3 sensors-23-07217-f003:**
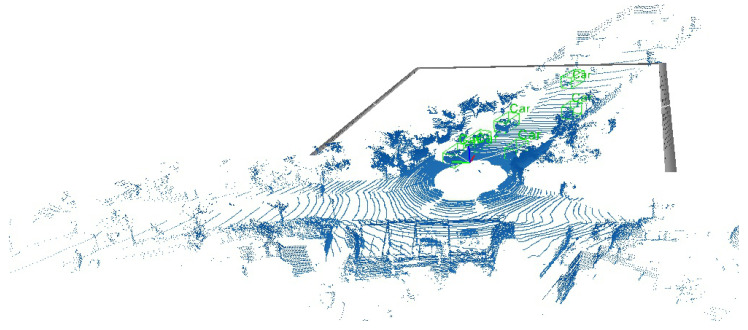
The 3D object detection qualitative results. Green boxes: vehicle. Black lines: front view.

**Table 1 sensors-23-07217-t001:** Performance comparison on the KITTI *val* split with $AP_{3D}$ calculated by 11 recall positions for the car detection. *: re-implemented with the official code.

Methods	${\mathrm{AP}}_{3D}$ (%)		
	Easy	Mod	Hard
PointRCNN [3]	88.88	78.63	77.38
VoxelNet [5]	81.97	65.46	62.85
Voxel R-CNN [1]	89.41	84.52	78.93
PointPillars [31]	86.62	76.06	68.91
SA-SSD [6]	90.15	79.91	78.78
VoxSeT [14]	89.21	86.71	78.56
SECOND [7]	87.43	76.48	69.10
PV-RCNN [4]	89.35	83.69	78.70
VoTr-SSD * [15]	87.86	78.27	76.93
**VoTr-DADA (SSD)**	88.57	79.31	77.74
VoTr-TSD * [15]	89.04	84.04	78.68
**VoTr-DADA (TSD)**	89.46	84.63	78.90

**Table 2 sensors-23-07217-t002:** Performance comparison on the KITTI test set with $AP_{3D}$ calculated by 40 recall positions for the car detection. *: re-implemented with the official code.

Methods	${\mathrm{AP}}_{3D}$ (%)		
	Easy	Mod	Hard
MV3D [32]	74.97	63.63	54
AVOD-FPN [33]	83.07	71.76	65.73
F-PointNet [2]	82.19	69.79	60.59
3D-CVF [9]	89.2	80.05	73.11
CLOCs [10]	88.94	80.67	77.15
VoxelNet [5]	77.47	65.11	57.73
SECOND [7]	83.34	72.55	65.82
PointPillars [31]	82.58	74.31	68.99
PointRCNN [3]	86.96	75.64	70.7
SA-SSD [6]	88.75	79.79	74.16
3DSSD [8]	88.36	79.57	74.55
PV-RCNN [4]	90.25	81.43	76.82
Voxel-RCNN [1]	87.95	79.71	75.09
CT3D [34]	87.83	81.77	77.16
VoxSeT [14]	88.53	82.06	77.46
VoTr-TSD * [15]	89.90	82.09	79.14
**VoTr-DADA (TSD)**	90.04	82.36	79.33

**Table 3 sensors-23-07217-t003:** Performance comparison on the Waymo Open Dataset with validation sequences for vehicle detection. *: re-implemented with the official code.

Methods	L1	L2	L1 mAP/mAPH by Distance		
	mAP/mAPH	mAP/mAPH	0–30 m	30–50 m	50 m–Inf
PointPillars [31]	63.3	55.2	84.9	59.2	35.8
MVF [35]	62.93	–	86.30	60.02	36.02
CVCNet [36]	65.2	–	86.80	62.19	29.27
Voxel-RCNN [1]	75.59	66.59	92.49	74.09	53.15
SECOND [7]	67.94	59.46	88.10	65.31	40.36
PV-RCNN [4]	70.3	65.4	91.9	69.2	42.2
VoxSeT [14]	76.02	68.16	91.13	75.75	54.23
VoTr-SSD * [15]	68.99	60.22	88.18	66.73	42.08
**VoTr-DADA(SSD)**	70.02	61.19	88.9	67.79	43.21
VoTr-TSD * [15]	74.95	65.91	92.28	73.36	51.09
**VoTr-DADA (TSD)**	75.69	66.74	92.46	73.83	51.61

**Table 4 sensors-23-07217-t004:** Comparisons of inference speeds for different frameworks on the KITTI dataset; 48 attending voxels are used.

Methods	Inference Speed (Hz)
**Vanilla VoTr**	**14.65**
VoTr with Conv-based Deformable Attention	12.08
VoTr with Density-Aware Deformable Attention	13.32

**Table 5 sensors-23-07217-t005:** Comparisons of the model parameters for different frameworks on the KITTI dataset.

Methods	Model Parameters
Vanilla VoTr	4.8 M
VoTr with Conv-based Deformable Attention	5.1 M
**VoTr with Density-Aware Deformable Attention**	**4.8 M**

**Table 6 sensors-23-07217-t006:** Effects of the number of attending voxels for each querying voxel on the KITTI *val* split.

Methods	Number of Attending Voxels	${\mathrm{AP}}_{3D}$ (%)
(a)	32 Voxels	78.19
(b)	40 Voxels	79.29
(c)	**48 Voxels**	**79.54**

**Table 7 sensors-23-07217-t007:** Comparisons of the different VoTr-DADA block structures on the KITTI *val* split; 48 attending voxels are used.

Method	Replaced DADA Blocks	${\mathrm{AP}}_{3D}$ (%)
(a)	Only First DADA Block	77.53
(b)	First and Second DADA Blocks	78.88
(c)	**Second and Third DADA Blocks**	**79.54**
(d)	All DADA Blocks	79.36

**Table 8 sensors-23-07217-t008:** Comparisons of the voxel search range parameter *r* to find the highest-density voxel around a querying voxel on the KITTI *val* split.

Method	Search Range Parameter *r*	${\mathrm{AP}}_{3D}$ (%)
(a)	2	78.87
(b)	**4**	**79.54**
(c)	8	77.61

## Data Availability

Not applicable.

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
