# Peer review of "Voxel Transformer with Density-Aware Deformable Attention for 3D Object Detection"

_sensors, 2023, doi:10.3390/s23167217_

Round 1
Reviewer 1 Report
This paper proposes a 3D object detection backbone VoTr-DADA, which improves VoTr by introducing a simple voxel density-based deformable attention mechanism (DADA). Experiment results verify the effectiveness of the proposed method.
There are some points that I am concerned as follows:
(1) The authors should increase an equation to show how the Epos takes part in the multi-head attention.
(2) In line 273-275, the authors declare that “For the hard car class, VoTr-DADA (TSD) achieves 78.90% mAP, outperforming all the previous methods”. However, Voxel R-CNN achieves a higher mAP with 78.93. Please explain it. Besides, please explain why the proposed method does not perform best for the Easy and Mod classes.
(3) In Table 1 and Table 2, the authors split the comparative methods into 3 and 2 groups, respectively, please describe the relationship in each group and compare the corresponding results to better demonstrate the performance of the proposed method.
Reviewer 2 Report
In this paper, the authors present the Voxel Transformer with Density-Aware Deformable Attention (VoTr-DADA). This innovative method for 3D object detection employs deformable attention guided by density. Combining the advantages of VoTr and Deformable Attention, this method effectively identifies critical regions within the input using density features while maintaining computational efficiency. The authors introduce the Density-Aware Deformable Attention (DADA) module in order to focus on relevant regions and flexibly capture more informative features. Experimental results on the KITTI dataset and the Waymo Open dataset show that this proposed method outperforms the baseline VoTr model in 3D object detection while maintaining a fast inference speed.
The topic of the paper is important from both theoretical and practical perspectives. The manuscript could be improved by considering the following points:
1. In the abstract you may state the novelty of the work. The abstract needs to be corrected with the required information. Less important lines can be removed. Make the abstract and proposal relevant.
2. Problem statement should be discussed in the first para of the Introduction part. Include the main objects of the work.
3. Related work must discuss the existing methods with their advantages and disadvantages. You can modulate the one para about existing limitations and proposed ideology
4. Architecture model (figure 1) is not clearly visualised and understandable. you could consider including model with good resolution. More explanation and discussion can be presented in this section.
5. Some equation terms must be ensured and defined before using the equation.
6. Dataset details should be discussed in the result section and what are the parameters used for experimental evaluation should discuss.
7. There are no advantages and a disadvantage discussed in the result section. There is no comparative analysis. Kindly include all the parts in the result analysis.
8. The quality of figures is less than normal, increasing the quality of figures.
9. Future scope of the article can be discussed with limitations in the conclusion part.
10. This paper needs rigorous revision in terms of techniques, English and presentation.
Moderate editing of English language required
Reviewer 3 Report
This paper developed density-aware deformable attention module exploits voxel densities to direct the attending voxels to more informative regions. Authors conducted experiments on KITTI and Waymo Open datasets. I have the following concerns:
1. The introduction is written well. Literature review also provides sufficient information about previous studies.
2. However, it would be nice if authors can provide related work section and a table to deeply analyze the methods used in previous studies and the drawbacks and advantages of similar studies.
3. I suggest to add the brief information about each section in the introduction section after main contributions.
4. Please provide the sample images of used datasets.
5. I suggest to add information and equations for to clarify used evaluation metrics.
6. Almost all information in experimental section written in text which is difficult to extract useful information. Thus, I suggest to add table for simple demonstrating dataset information and implementation details.
7. Please add other well know evaluation metrics such as precision and recall.
8. Please add visual representation of results.
9. I suggest to add discussion and future work section to show pros and cons of the proposed method.
English is fine.
Round 2
Reviewer 1 Report
The authors have answered my questions. I have no other comments.
Reviewer 2 Report
The revised manuscript has corrected all errors and is recommended for publishing.
Moderate editing of English language required
Reviewer 3 Report
The authors answered my comments and modified the manuscript.
English is fine.